# ARG1 mRNA Level Is a Promising Prognostic Marker in Head and Neck Squamous Cell Carcinomas

**DOI:** 10.3390/diagnostics11040628

**Published:** 2021-03-31

**Authors:** Barbora Pokrývková, Jana Šmahelová, Natálie Dalewská, Marek Grega, Ondřej Vencálek, Michal Šmahel, Jaroslav Nunvář, Jan Klozar, Ruth Tachezy

**Affiliations:** 1Department of Genetics and Microbiology, Faculty of Science, Charles University, BIOCEV, 252 50 Vestec, Czech Republic; jana.smahelova@natur.cuni.cz (J.Š); dalewska@hotmail.cz (N.D.); michal.smahel@natur.cuni.cz (M.Š.); jaroslav.nunvar@natur.cuni.cz (J.N.); 2Department of Pathology and Molecular Medicine, 2nd Faculty of Medicine, Charles University, 150 06 Prague, Czech Republic; Marek.Grega@fnmotol.cz; 3Department of Mathematical Analysis and Applications of Mathematics, Faculty of Science of the Palacky University in Olomouc, 771 46 Olomouc, Czech Republic; ondrej.vencalek@upol.cz; 4Department of Otorhinolaryngology and Head and Neck Surgery, 1st Faculty of Medicine, Charles University, University Hospital Motol, 150 06 Prague, Czech Republic; Jan.Klozar@fnmotol.cz

**Keywords:** head and neck carcinoma, arginase 1, HPV, prognosis, macrophages

## Abstract

Head and neck squamous cell carcinomas (HNSCC) can be induced by smoking or alcohol consumption, but a growing part of cases relate to a persistent high-risk papillomavirus (HPV) infection. Viral etiology has a beneficial impact on the prognosis, which may be explained by a specific immune response. Tumor associated macrophages (TAMs) represent the main immune population of the tumor microenvironment with a controversial influence on the prognosis. In this study, the level, phenotype, and spatial distribution of TAMs were evaluated, and the expression of TAM-associated markers was compared in HPV positive (HPV+) and HPV negative (HPV−) tumors. Seventy-three formalin and embedded in paraffin (FFPE) tumor specimens were examined using multispectral immunohistochemistry for the detection of TAM subpopulations in the tumor parenchyma and stroma. Moreover, the mRNA expression of TAM markers was evaluated using RT-qPCR. Results were compared with respect to tumor etiology, and the prognostic significance was evaluated. In HPV− tumors, we observed more pro-tumorigenic M2 in the stroma and a non-macrophage arginase 1 (ARG1)-expressing population in both compartments. Moreover, higher mRNA expression of M2 markers—cluster of differentiation 163 (CD163), ARG1, and prostaglandin-endoperoxide synthase 2 (PTGS2)—was detected in HPV− patients, and of M1 marker nitric oxide synthase 2 (NOS2) in HPV+ group. The expression of ARG1 mRNA was revealed as a negative prognostic factor for overall survival of HNSCC patients.

## 1. Introduction

Every year, more than 750,000 new patients are diagnosed worldwide with head and neck squamous cell carcinomas (HNSCC) [1]. Most cases relate to smoking and alcohol consumption, but a growing part of these tumors originates as a consequence of persistent infection with high-risk human papillomavirus (HR HPV) [2]. HNSCC tumors occur in a number of anatomical locations, but HPV positive (HPV+) tumors are mostly localized in the oropharynx [3]. Patients with HNSCC of viral etiology are younger and have a better prognosis, which may be explained by a specific immune response [2,3,4,5]. Currently, researchers are focusing on searching and defining new biomarkers for early disease detection and prognosis prediction as an addition to the classical tumor-node-metastasis (TNM) staging and histological grading.

The tumor microenvironment (TME) is a complex system of tumor cells and surrounding stroma, which originates from normal tissue cells, fibroblasts, endothelial cells, pericytes, immune cells, and extracellular matrix. As for the tumor cells, genetic alterations resulting in uncontrolled proliferation and phenotype changes are characteristic, the stromal cells remain genetically unchanged, but their activity is largely influenced by cytokines produced by tumor cells themselves or by other cells of the TME [6]. Tumor associated macrophages (TAMs) are the most abundant immune cells of the TME. Functionally, macrophages can be polarized into two extreme groups: classical M1 with pro-inflammatory and anti-tumorigenic functions and alternative M2, which act as anti-inflammatory and pro-tumorigenic. Additionally, M2 macrophages can be subdivided into M2a, M2b, M2c, and M2d phenotypes, linked to different inducers but sharing similar functional activities [7]. Between M1 and M2 polarizations, the phenotype plasticity was described, reflecting the complex cytokine environment in tumors. Thus, the categorization of TAMs into M1 and M2 is simplified [8,9]. During the early phase of tumorigenesis, macrophage precursors migrate into the TME in response to chemoattractants produced by tumor or stromal cells. The level of these chemoattractants, such as C–C motif chemokine ligand 2 (CCL2), colony-stimulating factor 1 (CSF1), or vascular endothelial growth factor (VEGF), is enhanced by the hypoxic environment [10,11]. After entering the TME, macrophage precursors polarize to the M1 phenotype upon the autocrine or paracrine stimulus with interferon gamma (IFN-γ) or other molecules, such as tumor necrosis factor alpha (TNF-α) and toll-like receptor (TLR) ligands. The M1 macrophages produce inflammatory cytokines, such as interleukin (IL)-1, IL-6, IL-12, TNF-α, and IFN-γ. The inflammatory environment generates reactive nitrogen and oxygen species leading to genome instability [12]. Consequently, macrophages acquire the M2 phenotype, which contributes to tumor progression [12,13]. For M2 polarization, mainly the presence of transforming growth factor beta (TGF-β) and IL-4, IL-10, and IL-13 seems crucial [7,13]. The role of M2 macrophages is mainly connected with VEGF, platelet derived growth factor (PDGF), fibroblast growth factor (FGF), and TGF-β production, as well as with matrix metalloproteinases (MMPs) secretion, all resulting in the increased migration and invasiveness of tumor cells. Moreover, by the secretion of CCL17, CCL22, and CCL24 by M2 TAMs, some other immune cells are attracted, such as Th2 lymphocytes, regulatory T cells, or basophiles, leading to the tumor tolerance [8]. As M2 TAMs were found to be connected with a worse prognosis in breast [14], gastric [15], rectal [16], and pancreatic carcinomas [17] as well as in HNSCC [18], the research is now focused on M2 TAMs detection and characterization in different tumor specimens. For TAMs detection, the pan-macrophage CD68 marker is used, but for characterization of the phenotype, additional markers are needed, e.g., CD80, CD11c, inducible nitric oxide synthase 2 (NOS2), or human leukocyte antigen–DR isotype (HLA–DR) for M1 and CD163, CD204, CD206, or arginase 1 (ARG1) for M2 [19,20,21]. The M1/M2 TAMs stratification also reflects the difference in the metabolism of the amino acid arginine, which has a huge impact on TAMs function in the TME. M1 TAMs predominantly metabolize L-arginine by NOS2 resulting in NO increase in the tissue and inhibition of proliferation, while M2 TAMs utilize arginine mainly via ARG1 to generate urea and ornithine. The ARG1 pathway potentiates proliferation and tissue repair and supports the tumor growth [22]. Because both metabolic pathways utilize the same substrate, the activity of one pathway is cross-inhibited by the other one [21].

The indoleamine-2,3-dioxygenase 1 (IDO1) can serve as another marker of the M2 phenotype [23]. IDO1 is an immunoregulatory enzyme degrading tryptophan to kynurenine, resulting in the local immunosuppressive environment and neovascularization. Continual expression of IDO1 is enabled by prostaglandin-endoperoxide synthase 2 (PTGS2; also known as cyclooxygenase-2, COX-2) [24], which is crucial for M2 TAMs polarization maintenance [25]. Increased IDO1 expression correlates with poor prognosis in patients with colorectal [26], breast [25], endometrial [27], and ovarian carcinomas [28]. Higher PTGS2 expression has been associated with worse prognosis in many cancers including HNSCC [29,30].

In this study, we analyzed the levels and distribution of M1 and M2 subpopulations of TAMs in different compartments of head and neck tumors of different etiology using multispectral fluorescent immunohistochemistry (fIHC). We also examined, in the corresponding samples of tumors, the mRNA levels of the selected TAMs markers IDO1, NOS2, PTGS2, CD163, and ARG1 by reverse transcription followed by quantitative polymerase chain reaction (RT-qPCR). Regardless of the tumor etiology, the stroma was always more infiltrated by macrophages of both phenotypes M1 and M2 than the tumor parenchyma. In comparison to HPV+ tumors, the stroma of HPV negative (HPV−) HNSCC was more infiltrated by the M2 TAMs. In these tumors, RNA expression of the M2 TAMs markers was also higher. An increased level of ARG1 mRNA and higher tumor stage were found as negative prognostic factors of patients with HNSCC.

## 2. Materials and Methods

### 2.1. Sample Collection, Processing, and Characterization

We analyzed 73 samples of primary HNSCC located in the oral cavity and oropharynx (ICD10: C01, C06, C09, C10). Samples were obtained within the study conducted in our laboratory in cooperation with the Department of Otorhinolaryngology and Head and Neck Surgery, First Faculty of Medicine, Charles University, and Motol University Hospital, Prague. All patients signed an informed consent and completed a questionnaire related to risk factors for HPV infection and HNSCC induction. The study was approved by the Ethical Committee of the Motol University Hospital. All samples were processed and checked by a pathologist immediately after surgery. Samples were classified using the TNM nomenclature (8th edition), and the Charlson comorbidity index was calculated (https://www.mdcalc.com/charlson-comorbidity-index-cci, accessed on 5 March 2021). One part of the tumor was fixed in 10% neutral formalin and embedded in paraffin (FFPE), and the other part was transported to the laboratory in RPMI medium (Sigma-Aldrich, St. Luis, MO, USA) at 4 °C. Tumor single cell suspension was isolated instantly as described previously [31] and was stored in RNA*later*^®^ Stabilization Solution (Life Technologies, Carlsbad, CA, USA) at −80 °C until processing. DNA and total RNA were isolated using the NucleoSpin RNA/DNA buffer set (Macherey-Nagel, Düren, Germany) according to the manufacturer’s instructions. The concentration of DNA and total RNA was measured by a NanoDrop 2000 Spectrophotometer (Thermo Fisher Scientific, Waltham, MA, USA), and the quality of RNA was checked by the Experion^™^ Automated Electrophoresis System (Bio-Rad, Hercules, CA, USA), both according to the manufacturer’s protocols. Until analysis, DNA was stored at −20 °C and RNA at −80 °C. The presence and type of HPV in samples was evaluated by PCR with broad spectrum GP5+/6+-5′ bio primers followed by reverse line blot analysis as specified before [32]. The active viral infection was determined by HPV E6 mRNA detection as described previously [33,34]. For one sample, DNA was not available, but the RNA analysis revealed the presence of HPV16 E6 mRNA.

### 2.2. Gene Expression Analysis

For relative quantification of the expression of selected TAMs markers (IDO1, PTGS2, NOS2, ARG1, and CD163), we performed RT-qPCR. Firstly, total RNA was treated by DNase and then reverse transcribed using M-MLV Reverse Transcriptase (both Promega, Madison, WI, USA), according to the manufacturer’s instructions. The qPCR reactions were performed in 10 µL volume of the reaction mixture in duplicates using Xceed qPCR SG 2x Mix Lo-ROX (IAB, Prague, Czech Republic), 0.4 µM primers, and 2 µL of 4 × diluted cDNA on a CFX96^™^ Real-Time System instrument (Bio-Rad, Hercules, CA, USA). The reaction conditions were as follows: 3 min at 95 °C followed by 40 cycles of 10 s at 95 °C and 30 s at 60 °C. All reactions were followed by a melting curve analysis. The *β-glucuronidase* (*GUS*) and *actin beta* (*ACTB*) genes were used as the reference genes for normalization. The sequences of primers are listed in Table 1. The obtained amplification plots were analyzed by Bio-Rad CFX Maestro (Bio-Rad, Hercules, CA, USA). The relative quantification was done by the GenEx^™^ v.6 software (MultiD Analyses AB, Gothenburg, Sweden) using the ∆∆Ct method.

### 2.3. Tissue Slides Preparation and fIHC Validation

From FFPE tumors, 2 µm sections were prepared on StarFrost Advanced Adhesive microscope slides (Knittel-Glass, Braunschweig, Germany). In the first and the last sections from paraffin blocks, the pathologist confirmed the tumor presence. Sections were stored at 4 °C and processed within three weeks from preparation. Slides were deparaffinized at 60 °C for two hours, brightened in xylene three times for 10 min, hydrated by alcohols of descending grades (all from Penta, Prague, Czech Republic), and fixed in 10% neutral buffered formalin (Diapath, Martinengo, Italy) for 20 min. Antigen retrieval (AR) was performed using microwave in AR9 Tris-EDTA buffer, pH 9.0 (Zytomed Systems, Berlin, Germany) or AR6 citrate buffer, pH 6.0 (Akoya Biosciences, Menlo Park, CA, USA) depending on the antibody used. Prior to a primary antibody incubation, the tissue was demarcated with an Elite Pap pen (Diagnostic BioSystems, Pleasanton, CA, USA) and blocked using Antibody Diluent/Block (Akoya Biosciences, Menlo Park, CA, USA) for 10 min at room temperature (RT). Primary antibodies against CD68 (clone KP1; Santa Cruz Biotechnology, Dallas, TX, USA), CD80 (62N3G8; Novus Biological, Centennial, CO, USA), CD163 (10D6; Thermo Fisher Scientific, Waltham, MA, USA), arginase 1 (A-2; Santa Cruz Biotechnology, Dallas, TX, USA), and Cytokeratin Pan Type I/II (AE1/AE3; Thermo Fisher Scientific, Waltham, MA, USA) were diluted in Antibody Diluent/Block (Akoya Biosciences, Menlo Park, CA, USA) and stained separately. Primary antibody incubation was followed by 10 min of staining with Opal Polymer HRP Ms+Rb, TSA-based Opal^™^ fluorophores, and DAPI counterstain, all from the Opal 7-Color Manual IHC Kit (Akoya Biosciences, Menlo Park, CA, USA). The stained tissue sections were mounted with Fluoromount^™^ Aqueous Mounting Medium (Sigma-Aldrich, St. Luis, MO, USA). Slides were imaged with a magnification of 20 × 10 using the Mantra Snap 2.0.0 software, and pictures were analyzed in the InForm 2.4.6 software (both Akoya Biosciences, Menlo Park, CA, USA). The reliability of the staining was confirmed by no primary and isotype controls, where the primary antibody was substituted with Antibody Diluent/Block or an appropriate isotype antibody, respectively.

### 2.4. Multiplex fIHC Staining

After monoplex staining validation, a panel for multiplexed staining was designed. The order of the stained antibodies was determined according to the epitope stability after multiple stripping rounds. Fluorophores have been assigned to antibodies to minimize spectral overlapping. Subsequent titration of antibodies and fluorophores was performed to obtain reliable staining with a fluorophore signal between 5 and 20 and a signal/noise ratio of >10. The order of antibody staining, reagent dilutions, and reaction conditions are listed in Table 2. For all antibodies in the panel, a stripping quality control was performed.

### 2.5. Multiplex fIHC Data Analysis

From each tumor sample, five regions of interest (ROI) were randomly selected across the tumor parenchyma and stromal area, and snapped using the Mantra Snap 1.0.3 software (Akoya Biosciences, Menlo Park, CA, USA), with a magnification of 20 × 10. Pictures were analyzed in batches using the InForm 2.4.6. software (Akoya Biosciences, Menlo Park, CA, USA) with the prepared algorithm. This algorithm consists of linear unmixing using the prepared spectral libraries and trainable steps of tissue segmentation, cell segmentation, and cell phenotyping. For tissue segmentation training, we marked the cytokeratin positive area as the tumor parenchyma, the cytokeratin-free area as the stroma, and the empty (DAPI-free) area as the background. Each step was optimized in different tumor pictures from the set of samples and then applied to the analyses of additional samples in the batches. We introduced six different cell phenotypes: M1 macrophages (M1; CD68+CD80+), M2 macrophages (M2; CD68+CD163+), M2 macrophages producing ARG1 (M2-ARG; CD68+ARG+), cells exclusive of macrophages producing ARG1 (ARG; ARG+), CD80+ only cells (CD80; CD80+), and the remaining cells negative for any macrophage marker (other). For the further analysis, the phenotypes “other” and “CD80” were not included as they serve only for phenotyping algorithm setting. For the final analysis, cells with the confidence of phenotyping higher than 75% were counted. According to our observation, this level was suitable for reliable phenotyping. For all the samples, the percentages of unsatisfactory cells ranged between 5% and 15%. The numbers of positive cells in the compartments were then calculated per megapixel (Mpx).

### 2.6. Statistical Analysis

All patients were grouped according to the HPV E6 mRNA positivity into HPV+ and HPV− groups. The levels of M1, M2, M2–ARG, and ARG phenotypes were expressed as cells per Mpx and were evaluated separately for the tumor parenchyma and stroma compartments in both sample groups. The differences in cell numbers/Mpx between groups were evaluated using the Mann–Whitney U test, and the differences between the tumor parenchyma and stroma parts in the corresponding patients were analyzed using the sign test. For RT-qPCR data analyses, the unpaired *T*-test was used to see the differences between groups, and the paired T test was performed for the comparison between compartments. To analyze the possible correlation between mRNA expression levels, the Pearson correlation coefficient was applied. The Cox proportional hazards model was used for the overall specific survival (OS) analysis. The Akaike information criterion (AIC) was used for model selection. The following demographic and clinical pathological factors were included: gender, age, education (≤12 years, >12 years), smoking (nonsmoker, ex-smoker, smoker), alcohol use (nondrinker, ex-drinker, drinker), location (oropharyngeal, oral), tumor extension (pT1-4), nodal status (pN0-3), tumor stage (S I, II, III, IV), metastasis (M 0, 1), extracapsular spreading (0, 1), Charlson comorbidity index, and HPV status (HPV E6 mRNA negative, HPV16 E6 mRNA positive). In addition to these factors, qPCR factors (mRNA expression of *IDO1*, *NOS2*, *PTGS2*, *CD163,* and *ARG1*) and fIHC factors (M1, M2, M2-ARG, and ARG cells/Mpx in both the tumor parenchyma and stroma) were tested. Tumor extension, nodal status, tumor stage, and histological grading were numeric measures. As an addition to the Cox model, plots for Kaplan–Meier estimator were created.

A *p* value of < 0.05 was considered as a significant difference. All statistical analyses were performed using the GraphPad Prism 8.4.2 software (GraphPad Software, San Diego, CA, USA) and R version 4.0.2 (https://www.R-project.org/, accessed on 5 March 2021) [35].

## 3. Results

### 3.1. Patients Characterization

For this study, a total of 73 patient samples of primary HNSCC were collected and characterized. The median follow-up period was 18.5 months. HPV DNA was found in 46/72 (64%) samples. Of 46 HPV DNA positive samples, 44 (96%) were HPV16 DNA positive, one (2%) sample was positive for HPV16 and HPV53, and one (2%) sample was positive for HPV35 DNA. We divided the samples according to the presence of viral E6 mRNA, a marker of active viral infection, into HPV+ (38/73, 52%) and HPV− (35/73, 48%). In 10 cases of 35 HPV− tumors (29%), only HPV DNA and no E6 mRNA was detected pointing to non-active viral infection. All 38 HPV+ tumors were located in the oropharynx, unlike the HPV− tumors, with the majority of samples (23/35, 66%) obtained from the oral cavity and the remaining samples (12/35, 34%) from the oropharynx. The age of patients ranged from 41 to 83 years, with a median of 60 years for HPV+ patients and of 62.5 for HPV− cases. Consistently with previous observations [33,36], the tumors were mostly collected from men (57/73, 78%). Detailed patient and tumor characteristics are listed in Table 3.

### 3.2. Expression of M2 TAM Markers Is Higher in HPV− Tumors

We examined the mRNA level of TAMs associated genes—*ARG1*, *CD163*, *NOS2*, *IDO1,* and *PTGS2*—in HPV+ and HPV− tumors. For this purpose, we performed RT-qPCR analysis with relative quantification by the ΔΔCt approach using *GUS* and *ACTB* as the reference genes in 69 samples (38 HPV+ and 31 HPV−). Remaining four samples of the cohort were excluded due to insufficient RNA quality. We detected significantly higher levels of CD163 (*p* = 0.0035), ARG1 (*p* = 0.0009), and PTGS2 (*p* = 0.0402) mRNA in HPV− tumors (Figure 1). These genes are considered as M2 associated markers. Next, NOS2 mRNA, the M1 marker, was detected in higher level in HPV+ tumors (*p* = 0.0103). On the contrary, we did not observe any differences in the IDO1 mRNA level between HPV+ and HPV− tumors.

These results show higher expression of the M2 markers *ARG1*, *CD163* together with *PTGS2*, which directly supports the M2 TAMs polarization and maintenance, in HPV− patients. To analyze the relationships among TAMs markers, we performed correlation analysis of mRNA levels by computing the Pearson correlation coefficient. As demonstrated in Figure 2, we observed a positive association between CD163 and ARG1 mRNA (*r* = 0.3179, *p* = 0.0078) and ARG1 and PTGS2 mRNA (*r* = 0.251, *p* = 0.0375). We also detected weak inverse correlation between ARG1 and NOS2 mRNA (*r* = −0.2824 *p* = 0.0187) which suggests an exclusive activity of ARG1 and NOS2 enzymes in M2 and M1 TAMs, respectively. Next, we observed weak association between IDO1 and NOS2 mRNA (*r* = 0.253, *p* = 0.0359), which points to a co-expression of these enzymes in inflammatory conditions [23].

### 3.3. The Stroma of HNSCC Is More Infiltrated by TAMs Than the Tumor Parenchyma

We introduced a panel of six antibodies, which can detect and quantify M1 and M2 macrophages in the tumor parenchyma and surrounding stroma (Appendix A). A total of 73 tumor specimens were stained using multiplex fIHC and analyzed by an algorithm for cell and tissue segmentation and cell phenotyping. This analysis revealed that the stroma was more infiltrated by TAMs of both phenotypes, M1 and M2, than the tumor parenchyma in both HPV+ and HPV− patients (Table 4).

### 3.4. The Numbers of Arginase 1 Positive Cells Are Higher in Both Tumor Parenchyma and Stroma of HPV− Patients

We detected more arginase 1 producing M2 TAMs (M2-ARG) in the stroma of HPV− patients (Figure 3A; *p* = 0.003) while the level of M2-ARG in the tumor parenchyma was comparable in both groups (*p* = 0.0843). The levels of the second M2 population (CD68+CD163+) were similar in both groups (*p* = 0.1869 for tumor parenchyma and *p* = 0.7626 for stroma). Similar levels of M1 macrophages were also found (*p* = 0.5309 and *p* = 0.6884, respectively). Next, we observed significantly higher infiltration by cells expressing only ARG1 both in the tumor parenchyma and stroma of HPV− patients (Figure 3B, *p* = 0.0092 and *p* = 0.0002, respectively). According to phenotype assignment, these cells are not considered TAMs, but other ARG1 expressing cells, such as neutrophils or myeloid-derived suppressor cells (MDSCs). Representative staining of M2 and M2-ARG cells is shown in Appendix A.

### 3.5. ARG1 Expression Level, HPV Status, and Tumor Stage Are Factors Influencing OS

Cox proportional hazards models were used for the OS analysis. In the univariate model, HPV positivity was observed as a strong predictor of survival (HR = 0.050, *p* = 0.0045). The best model selected by the Akaike information criterion (AIC) contained three predictors, HPV status, tumor stage, and ARG1 mRNA level. According to the results, OS of patients was negatively influenced by higher ARG1 mRNA level (HR = 1.4872, *p* = 0.0034) and increasing tumor stage (HR = 1.9602, *p* = 0.0246). In this model, HPV status was not significantly associated with the OS (HR = 0.2227, *p* = 0.2116). We observed that the majority of HPV+ patients have a lower tumor stage (*p* < 0.0001 by Fisher test) and lower ARG1 mRNA level compared to HPV− patients (*p* = 0.0009, Figure 1). Additionally, a higher ARG1 mRNA level and tumor stage (S = 4) in the HPV− cohort suggested a high risk for these patients (Figure 4). As an addition to the Cox model, Kaplan–Meier estimator plots for HPV status, tumor stage, and ARG1 mRNA level were created (Appendix A).

These results confirm the HPV status as the predictor of survival in a univariate analysis. In the Cox model including additionally ARG1 mRNA level and tumor stage, the effect of HPV positivity to OS was not significant most likely due to the correlation of higher tumor stage and HPV negativity.

## 4. Discussion

In this study, we detected ARG1 expression and higher pathological stage of the tumor as a negative prognostic factor for overall survival in patients with head and neck tumors. In HPV− tumors, higher expression of markers of the pro-tumorigenic M2 macrophages *ARG1, CD163,* and *PTGS2* was detected. These results were supported by the detection of higher levels of M2 TAMs by multispectral fIHC. Using the fIHC method, higher numbers of stromal M2 macrophages evidenced by the presence of ARG1 protein were detected in the HPV− cohort. Furthermore, in both parenchyma and stroma of tumors from the HPV− patients, we have also detected higher numbers of non-macrophage populations (ARG phenotype), represented predominantly by MDSCs or neutrophils [37,38,39,40].

The detection and detailed phenotyping of cells in the TME is enabled by the introduction of advanced multiplex immunohistochemical methods. Multispectral immunohistochemistry is a powerful tool, which allows for analyzes of immune cells in situ with respect to the localization in different tumor compartments. The stratification into the tumor parenchyma and surrounding stroma helps to understand the relationships among cells in the TME and reveals the possible influence of the type and number of immune cells in different compartments on patients’ prognosis. Such approach is a basis for defining the immunoscore. This indicator, which is already routinely used for the estimation of recurrence risk in colon cancer patients, is independent of the TNM staging [41]. For HNSCC, the tumor classification using the TNM methodology is predictive of clinical outcomes. It has recently been modified for the oropharyngeal tumors by introducing an indirect marker for HPV status, overexpression of the p16 protein, which identifies active HPV infection [3,42]. It has been shown—by us and others—that tumor etiology is an important predictor of clinical outcome in HNSCC patients [32,33,43]. In this study, we confirmed the HPV status as the strongest marker of better prognosis of these patients. However, the better prognosis of patients with HNSCC of viral etiology is still a matter of intensive research, and there is evidence that the response of the immune system varies with the etiology of tumors [44,45]. Therefore, a more accurate prediction of the prognosis and treatment response can be achieved by introducing the immunoscore into the staging system. Numerous reports analyzed types and numbers of immune cells in HNSCC tumors, but there were important differences in the methodological approaches [44,45,46,47].

In our study, we focused on the detection and characterization of TAMs as the main immune population of TME. The influence of TAMs on the prognosis or disease progression has previously been described in HNSCC with controversial conclusions [45,46,47,48,49]. In those studies, macrophages were only detected based on the CD68 marker, which does not allow for their detailed phenotyping and evaluation of different roles in TME. Moreover, most studies did not address the HPV status. In our study, additional markers were used for a better stratification of macrophage phenotypes and were correlated with HPV status. We showed that the stroma in HNSCC was more infiltrated by both M1 and M2 (M2 and M2-ARG) macrophages than the tumor parenchyma, regardless of HPV status. In the HPV− cohort, higher abundance of M2 macrophages producing ARG1 (M2-ARG) was detected in the stroma compared to the HPV+ patients. In contrast, the M1 and M2-CD163+ populations were equally present in HPV+ and HPV− tumors in both compartments. A similar study of Ou et al. only used the CD163 marker for the identification of M2 TAMs. Based on the CD68+ M1 to CD68+CD163+ M2 ratio, they showed higher abundance of CD68+CD163+ M2 TAMs in the stroma of HPV− HNSCC [50].

The localization of TAMs in the TME is crucial. In breast carcinoma, higher infiltration of CD163+ macrophages in the stroma correlated with worse OS [14], but higher CD163+ TAMs infiltration in tumor invasive front was correlated to an improved prognosis of colorectal carcinoma patients [51]. In a recent study using the multispectral immunohistochemistry approach, the stroma of colorectal carcinoma was predominantly infiltrated by M2 TAMs, but no effect on patients’ survival was observed [52]. Similarly, using the fIHC method, higher infiltration with M2 macrophages (CD68+CD163+CD206+) in the stroma was found in gastric carcinoma, and, conversely, higher M1 populations were detected in a tumor-nest area [53]. Higher CD163+ TAMs infiltration in the stroma compared to the tumor parenchyma was also reported in esophageal carcinoma where it correlated with worse OS, but an impact of elevated CD163+ TAMs infiltration on tumor aggressiveness was observed in both compartments [54]. In HNSCC, higher infiltration with CD163+ macrophages in the stroma was related to worse survival of patients [18,55], but when the HPV status was included in the model, no prognostic effect of CD163+ cells on OS was evidenced. For progression-free survival (PFS), the presence of CD163+ cells in HPV− tumors was a negative prognostic factor [55]. The number of CD163+ TAMs increased with tumor grade in oral squamous cell carcinoma. Furthermore, in low-grade carcinomas, CD163+ TAMs were mainly located in the stroma, and the number of CD163+ TAMs in the tumor nest also increased with tumor grade [56].

Our fIHC data were further supplemented with mRNA analysis of TAM markers using RT-qPCR. In the HPV− cohort, higher expression levels of the M2 TAMs genes *ARG1*, *CD163,* and *PTGS2* were detected, while NOS2 mRNA, the M1 marker, was more expressed in tumors of HPV+ patients. Taken together, these results suggest a higher prevalence of M2 TAMs in the TME of non-virally induced HNSCC. In the study by Ohashi et al., predominant abundance of M2 TAMs was detected in the HNSCC by qPCR measurement of CD68, CD163, and CSF1 receptor (CSF1R) mRNA, but the HPV status of tumors was not examined. They also observed higher mRNA level of TAM markers in HNSCC compared to the healthy pharyngeal tissue [57]. PTGS2 is a well-characterized enzyme whose upregulation in HNSCC has already been described [30,44,58]. Higher expression of the *PTGS2* gene in HPV− tumors is in agreement with previously published data [44]. We further observed the association between the expression levels of CD163 and ARG1 or ARG1 and PTGS2 markers of M2 TAMs. Moreover, we detected negative association between ARG1 and NOS2 mRNA, which points to an exclusivity of corresponding enzymes activities in M2 and M1 macrophages, respectively. It was described previously, that activities of ARG and NOS2 enzymes are cross-inhibited resulting in different arginase metabolism in M1 and M2 TAMs [21]. Furthermore, we observed weak association between IDO1 and NOS2 mRNA. Our results are in agreement with the recent study of Wang et al., who observed correlation of IDO1 and NOS2 mRNA in pancreatic carcinoma [59] and with the study of Soliman et al., who observed positive correlation of IDO1 and NOS2 on the protein level on breast cancer tissue sections [60]. It was described that IDO1 and NOS2 are co-expressed as a result of infection or inflammation in human tissues. Moreover, NOS2-mediated NO increase results in inhibition of IDO1 activity in human tissues [23], but Wang et al. proposed the opposite mechanism where increasing NO potentiates the IDO1 activity [59]. The IDO1 activity in tumorigenesis is a complex system, which needs further clarification.

Besides the TAMs analysis, we also detected the non-macrophage cells producing ARG1 in the TME using fIHC. In addition to M2 TAMs, ARG1 is an accepted marker for MDSCs in gastric carcinoma [38], non-small cell lung carcinoma [39], and pancreatic adenocarcinoma [40] or a marker of neutrophils [61]. Here, we detected non-macrophage ARG1 expressing populations in higher numbers in both tumor parenchyma and stroma of HPV− tumors. *ARG1* expression was described as a factor promoting tumor growth [22]. The increased amount of the ARG1 protein in TAM or non-macrophage populations of HPV− tumors was also reflected by an increase in ARG1 mRNA. In HPV− patients, higher level of ARG1 mRNA was detected by RT-qPCR. Interestingly, a higher ARG1 mRNA level was associated with worse OS of HNSCC patients in our study. To our knowledge, our results are the first to show the impact of ARG1 mRNA expression on the prognosis of HNSCC. Similar results were observed in patients with colorectal cancer, where a higher ARG1 mRNA level was associated with worse OS and disease-free survival (DFS) [62], or classic Hodgkin lymphoma, where an elevated level of ARG1 negatively influenced PFS [37]. The overexpression of ARG1 in HNSCC tissue and peripheral blood compared to healthy donors was described by Shrivastava et al. [63], but opposite results were shown by Ohashi et al. [57]. Both studies did not address tumor etiology or prognosis, which may explain the discrepancy of the results.

## 5. Conclusions

In this study, the levels and distribution of M1 and M2 TAMs were analyzed in the TME of HNSCC, and expression of TAM-associated markers was evaluated with respect to HPV status. Regardless of the tumor etiology, higher levels of both M1 and M2 TAMs were detected in the stroma. In HPV− tumors, the stroma was more infiltrated by M2 TAMs expressing ARG1 compared to HPV+ tumors. Moreover, expression of the M2 TAM-associated markers CD163, ARG1, and PTGS2, was higher in HPV− tumors. In HPV− patients, higher non-macrophage populations expressing ARG1 were detected in both compartments. A higher level of ARG1 mRNA was found to be a new negative prognostic factor for OS of patients with HNSCC.

## Figures and Tables

**Figure 1 diagnostics-11-00628-f001:**
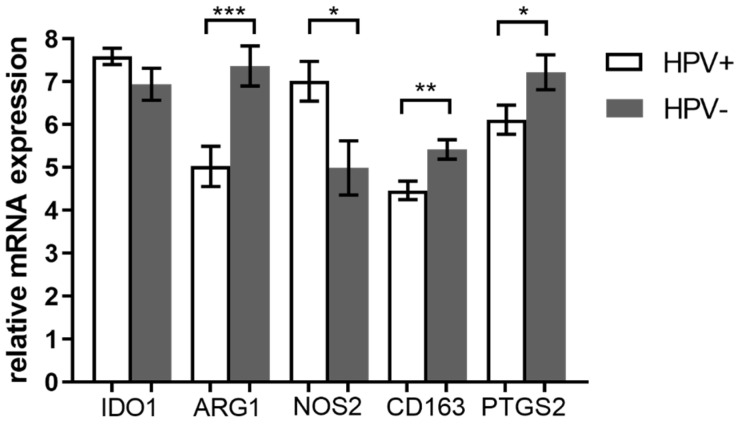
Comparison of gene expression in HPV+ and HPV− tumors. Relative mRNA expression of M1 and M2 tumor associated macrophage (TAM)-associated markers were compared in tumors with different etiology. Measured mRNA levels were normalized to *GUS* and *ACTB* reference genes using ΔΔCt approach. Significantly higher arginase 1 (ARG1), CD163, and prostaglandin-endoperoxide synthase 2 (PTGS2) mRNA levels were detected in the HPV− cohort (*** *p* = 0.0009 for ARG1, ** *p* = 0.0035 for CD163, and * *p* = 0.0402 for PTGS2 by unpaired *t*-test) and higher nitric oxide synthase 2 (NOS2) mRNA level was detected in the HPV+ cohort (* *p* = 0.0103 by unpaired *t*-test). Error bars—standard error of the mean.

**Figure 2 diagnostics-11-00628-f002:**
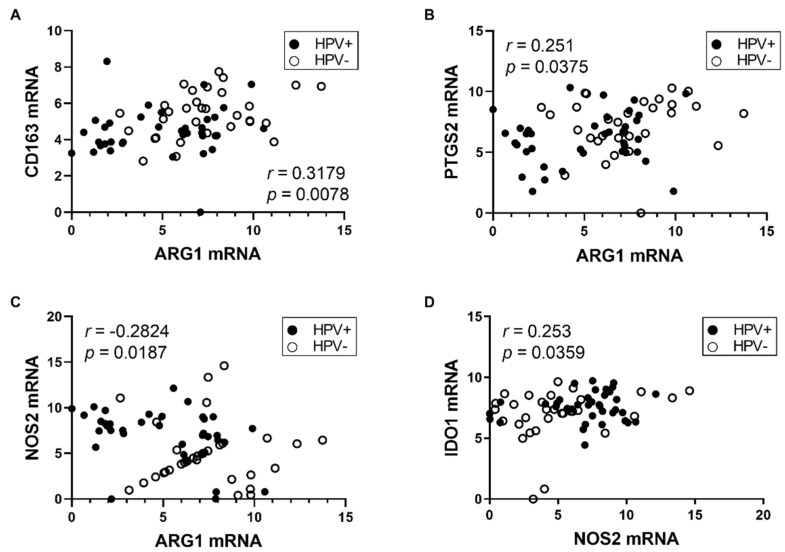
Correlation of relative gene expressions. Pearson correlation coefficient was measured between (**A**) ARG1 and CD163, (**B**) ARG1 and PTGS2, (**C**) ARG1 and NOS2, and (**D**) NOS2 and IDO1 relative mRNA levels.

**Figure 3 diagnostics-11-00628-f003:**
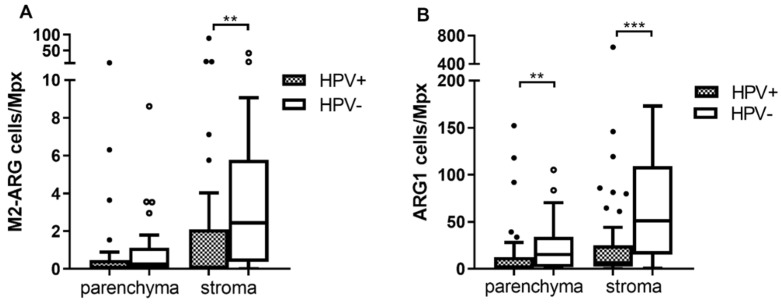
Multispectral fIHC analysis. Numbers of analyzed cells/Mpx in tumor parenchyma and stroma were compared between HPV+ and HPV− tumors using Mann–Whitney U test. (**A**) Number of M2 macrophages expressing ARG1 (M2-ARG)/Mpx was higher in stroma of HPV− patients (** *p* = 0.003), in contrast to tumor parenchyma where the difference was not significant; (**B**) higher number of ARG1 cells/Mpx was observed in the HPV− patients both in tumor parenchyma (** *p* = 0.0092) and stroma (*** *p* = 0.0002).

**Figure 4 diagnostics-11-00628-f004:**
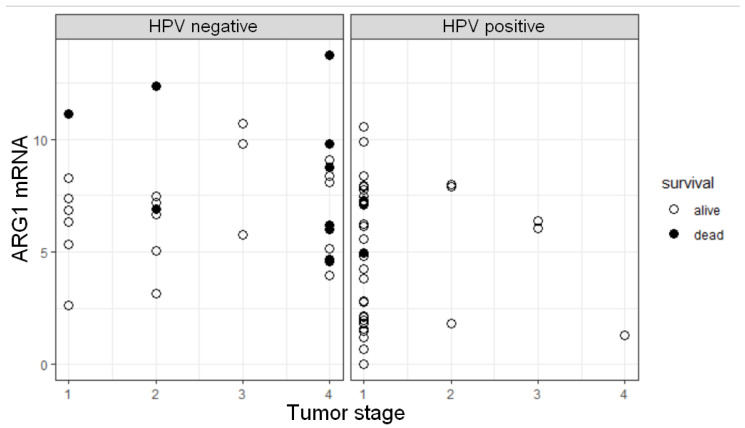
Relationships between ARG1 mRNA expression, tumor pathological stage, and HPV status as the predictors of overall survival. Each point reflects separate observation of a given predictor value.

**Table 1 diagnostics-11-00628-t001:** Sequences of primers with expected product lengths.

Gene		Sequence 5′–3′	Length [bp]
*IDO1*	F	AAGAAACTGGAACTGCCTCCT	121
	R	CACGAAATGAGAACAAAACGTCC	
*ARG1*	F	GGCAGAAGTCAAGAAGAACGGA	127
	R	GTGAGCATCCACCCAGATGA	
*CD163*	F	GCAATGGGGTGGACTTACCT	120
	R	TGCTTCACTTCAACACGTCC	
*NOS2*	F	GCTGTGCTCCATAGTTTCCAG	137
	R	GGGACCAGCCAAATCCAGTC	
*PTGS2*	F	GCATTCTTTGCCCAGCACTT	142
	R	GGCGCAGTTTACGCTGTCT	
*GUS*	F	GAAAATATGTGGTTGGAGAGCTCATT	101
	R	CCGAGTGAAGATCCCCTTTTTA	
*ACTB*	F	CCACGAAACTACCTTCAACTCCA	132
	R	GTGATCTCCTTCTGCATCCTGTC	

F—forward; R—reverse; bp—base pairs.

**Table 2 diagnostics-11-00628-t002:** Design of the panel for multiplex fluorescent immunohistochemistry (fIHC) staining.

#	AR	Antibody	Dilution	Incubation	Secondary	OPAL	Dilution
1	6	CD68	1:200	1 h/RT	Ms + Rb Opal HRP polymer	540	1:200
2	6	CD163	1:100	OVN/4 °C		620	1:100
3	9	ARG1	1:200	1 h/RT		650	1:200
4	9	CD80	1:50	OVN/4 °C		520	1:50
5	6	Cytokeratin	1:800	1 h/RT		690	1:200
6	6	DAPI	1:15	5 min/RT	–	–	–

AR—antigen retrieval buffer; RT—room temperature; OVN—overnight.

**Table 3 diagnostics-11-00628-t003:** Demographic and clinical characterization of the cohort.

Characteristics		Total	HPV+	HPV−
		No. (%)	No. (%)	No. (%)
No. of patients		73 (100%)	38 (52%)	35 (48%)
Age (years)	Mean age	61.86	61.66	62.14
	Median age	61.50	60.00	62.50
Gender	female	16 (22%)	3 (8%)	13 (37%)
	male	57 (78%)	35 (92%)	22 (63%)
Localization	oropharynx	50 (68%)	38 (100%)	12 (34%)
	oral cavity	23 (32%)	0 (0%)	23 (66%)
Education	>12 years	31 (50%)	19 (54%)	12 (44%)
	≤12 years	31 (50%)	16 (46%)	15 (56%)
Smoking	never	24 (33%)	19 (50%)	5 (14%)
	past	23 (31%)	12 (32%)	11 (32%)
	current	26 (36%)	7 (18%)	19 (54%)
Alcohol consumption	never	20 (27%)	14 (37%)	6 (17%)
	past	11 (15%)	2 (5%)	9 (26%)
	current	42 (58%)	22 (58%)	20 (57%)
Tumor extension (pT)	T1	17 (23%)	7 (18%)	10 (29%)
	T2	48 (66%)	29 (76%)	19 (54%)
	T3	4 (5.5%)	1 (3%)	3 (8.5%)
	T4	4 (5.5%)	1 (3%)	3 (8.5%)
Nodal status (pN)	N0	30 (41%)	10 (29%)	20 (57%)
	N1	32 (44%)	26 (68%)	6 (17%)
	N2	5 (7%)	1 (1.5%)	4 (12%)
	N3	6 (8%)	1 (1.5%)	5 (14%)
Metastasis presence (M)	0	69 (95%)	38 (100%)	31(89%)
	1	4 (5%)	0 (0%)	4 (11%)
Tumor stage (S)	I	40 (55%)	32 (84%)	8 (23%)
	II	12 (16%)	3 (8%)	9 (26%)
	III	6 (8%)	2 (5%)	4 (11%)
	IV	15 (21%)	1 (3%)	14 (40%)
Extracapsular spreading	0	57 (78%)	28 (74%)	29 (83%)
	1	16 (22%)	10 (26%)	6 (17%)
Charlson comorbidity index	0	2 (3%)	2 (5%)	0 (0%)
	1	19 (26%)	13 (34%)	6 (17%)
	2	16 (22%)	7 (18.5%)	9 (26%)
	3	16 (22%)	7 (18.5%)	9 (26%)
	4	10 (13%)	6 (16%)	4 (11%)
	5	5 (7%)	1 (3%)	4 (11%)
	6	2 (3%)	0 (0%)	2 (6%)
	7	3 (4%)	2 (5%)	1 (3%)

**Table 4 diagnostics-11-00628-t004:** TAMs and arginase (ARG) phenotype infiltration in tumor parenchyma and stroma expressed by median values of cells/Mpx.

	HPV+			HPV−			All Patients		
Phenotype	*Parenchyma*	*Stroma*	*p*	*Parenchyma*	*Stroma*	*p*	*Parenchyma*	*Stroma*	*p*
M1	1.635	29.68	<0.0001	4.693	22.3	0.0001	2.361	26.9	<0.0001
M2	0.982	17.6	<0.0001	0.472	11.87	<0.0001	0.789	12.8	<0.0001
M2-ARG	0	0	0.0347	0.255	2.442	<0.0001	0.203	1.007	<0.0001
ARG	1.462	6.314	<0.0001	15.1	51.12	<0.0001	4.023	18.13	<0.0001

## Data Availability

The data presented in this study are available in Appendix A.

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
