# Peer review of "ARG1 mRNA Level Is a Promising Prognostic Marker in Head and Neck Squamous Cell Carcinomas"

_diagnostics, 2021, doi:10.3390/diagnostics11040628_

Round 1
Reviewer 1 Report
This is an interesting report focusing on tumor associated macrophages (TAMs) and demonstrating that ARG1 mRNA level may serve as a promising prognostic marker in head and neck squamous cell carcinomas (HNSCC).
The authors should consider the following points in order to improve the quality of the presentation of their findings:
1) In the abstract, the authors use the acronym HNC for head and neck carcinomas. However, the 73 patients investigated in the present study were all diagnosed with head and neck squamous cell carcinomas (HNSCC), and the acronym HNSCC has been used throughout the article. Please use HNSCC also in the abstract.
2) Carefully check the text for misspellings. For in stance, see line 288: "We also detected week inverse correlation between...", "week" should be "weak".
3) It is of concern that the authors have performed multiplex fIHC staining, but they do not provide representative pictures for any of the staining. A few representative pictures, at least for the most important findings (i.e., those referred to in Figure 3) should be included. Please add at least representative pictures of ARG1 cells and M2-ARG cells in both the parenchyma and stroma of HPV+ and HPV- HNSCC cases.
4) Figures 3A and 3B are not adequately referred to in the text.
5) In Figure 2, it seems that the authors have assessed the various correlations between mRNA markers by pooling all samples (HPV+ and HPV- HNSCC cases). What about these correlations if HPV+ and HPV- HNSCC cases are analyzed separately?
Author Response
1) In the abstract, the authors use the acronym HNC for head and neck carcinomas. However, the 73 patients investigated in the present study were all diagnosed with head and neck squamous cell carcinomas (HNSCC), and the acronym HNSCC has been used throughout the article. Please use HNSCC also in the abstract.
We thank the reviewer for pointing this out. We have corrected the abbreviation in the abstract (at line 18 and 32).
2) Carefully check the text for misspellings. For instance, see line 288: "We also detected week inverse correlation between...","week" should be "weak".
We thank the reviewer for his comment. The whole text was checked again for misspelling.
3) It is of concern that the authors have performed multiplex fIHC staining, but they do not provide representative pictures for any of the staining. A few representative pictures, at least for the most important findings (i.e., those referred to in Figure 3) should be included. Please add at least representative pictures of ARG1cells and M2-ARG cells in both the parenchyma and stroma of HPV+ and HPV- HNSCC cases.
We thank the reviewer for this suggestion. We added the representative pictures of IHC staining to the Supplementary files Figure S1 and S2.
4) Figures 3A and 3B are not adequately referred to in the text.
We have corrected the references of Figures 3A and 3B in the main text.
5) In Figure 2, it seems that the authors have assessed the various correlations between mRNA markers by pooling all samples (HPV+ and HPV- HNSCC cases). What about these correlations if HPV+ and HPV- HNSCC cases are analyzed separately?
We thank the reviewer for his comment. All samples were also analysed separately according to the HPV status. Except of ARG1-NOS2, we observed similar values of correlation coefficients in the HPV+ and HPV- groups, but these values were not statistically significant (due to fewer observations). We observed significant negative ARG1-NOS2 correlation in the HPV+ group, but not in the HPV- group.
|
All samples |
HPV- |
HPV+ |
|||
cor |
p-val |
cor |
p-val |
cor |
p-val |
|
ARG1-NOS2 |
-0.2824 |
0.0187 |
0.0085 |
0.964 |
-0.3643 |
0.0245 |
ARG1-CD163 |
0.3179 |
0.0078 |
0.3534 |
0.0512 |
0.1131 |
0.4989 |
ARG1-PTGS2 |
0.251 |
0.0375 |
0.1562 |
0.4015 |
0.187 |
0.2608 |
IDO1-NOS2 |
0.253 |
0.0359 |
0.2435 |
0.1869 |
0.1502 |
0.368 |

Reviewer 2 Report
This is an interesting paper deserving publication. I only suggest authors to provide kaplan-meier analysis graph as well as table with cox regression analysis of the survival.Author Response
This is an interesting paper deserving publication. I only suggest authors to provide kaplan-meier analysis graph as well as table with cox regression analysis of the survival.
We thank the reviewer for his suggestion. We provided the Kaplan-Meier graphs on the Supplementary Figure S3.

Reviewer 3 Report
The authors present a nicely written and structured paper on ARG1 mRNA being of prognostic use in HNSCC. For this purpose 73 HNSCC tissue samples were analysed from oral cavity (1/3) and oropharynx (2/3). It focuses mainly on TAMs, tumor stage and HPV-status. Several markers for different macrophages subpopulations were measured by PCR and IHC. Overall the results are convincing.
I would have following comments and questions to the authors:
Maybe a comparison to the healthy tissue could have been done.
The follow-up period for survival is rather short with 18.5 and can be a serious limitation of the study. Maybe the progression free survival would be a better measure or could be added to the results.
Looking into the human protein atlas online (https://www.proteinatlas.org/ENSG00000118520-ARG1/pathology/head+and+neck+cancer) the overall survival seems to be better with high ARG1 expression. This seems to be contradictory to the presented results.Please discuss.
Another thing that is briefly mentioned by the authors is that the 2 groups HPV+ and HPVneg are somewhat diverging concerning their corresponding tumor stages. Did the authors see different results considering the TNM atlas 7th edition that is currently still used to treat the patients?
The paper by Ohashi et al. 2017 https://doi.org/10.1111/cas.13244 should be cited, since some results go in pairs : M2 macrophages, CD163 mRNA expression levels, HNSCC, ARG1. Please discuss.
In this paper The following statement is made in reference to their results and the citation to ref. 18. “HNSCC tissue had a significantly lower expression of ARG1 than that in normal pharyngeal tissue (Fig. S2). This result suggests that ARG1 is not an M2 macrophage marker in human tumor tissue.18“ Please comment on this.
A minor point is that in table 3, tumor size is indicated, but I assume the T stage from TNM is meant. In this case this is not correct, since t-stage may be dependent on infiltration on surrounding structures and not only size. Tumor extension for example would be more adequate.
Did the authors checked the ARG1 for the group that is HPV positive and (ex-)smoker? What were the results?
Author Response
Maybe a comparison to the healthy tissue could have been done.
We thank the reviewer for his comment. However, our study was focused on comparison of TAMs level in the microenvironment of HNSCC of different etiology and on the evaluation of impact of different TAM infiltration on patients’ prognosis.
The follow-up period for survival is rather short with 18.5 and can be a serious limitation of the study. Maybe the progression free survival would be a better measure or could be added to the results.
We thank the reviewer for his suggestion. Besides the OS analysis, available data allow to analyse Disease Specific Survival. However, we did not perform this kind of analysis due to the low number of deaths from HNSCC.
Looking into the human protein atlas online (https://www.proteinatlas.org/ENSG00000118520-ARG1/pathology/head+and+neck+cancer) the overall survival seems to be better with high ARG1 expression. This seems to be contradictory to the presented results. Please discuss.
We thank the reviewer for his comment. We checked the data on human protein atlas and compared them with our results. In our cohort, the tumor stages I to III are predominant, unlike the cohort listed on human protein atlas. If the same criteria (tumor stage I to III and cut off value 0.62) are set, the high ARG1 expression relates to the worse survival, which agrees with our data.
Another thing that is briefly mentioned by the authors is that the 2 groups HPV+ and HPVneg are somewhat diverging concerning their corresponding tumor stages. Did the authors see different results considering the TNM atlas 7th edition that is currently still used to treat the patients?
The group of patients includes partially the time period when TNM7 was in use. The majority of cases was classified according to TNM8. In order to create a homogenous classification, we re-classified the few cases from the older period according to TNM8, which is currently used in clinical decision making. The reviewer might find strange that the nodal status of one patient from the HPV positive group was N3. This irregularity is due to the fact that p16 positivity is determinative for TNM classification while in our paper the presence of E6 mRNA, a much more specific marker, was used for determination of HPV status. So, in our opinion, this just demonstrates the imperfection of p16 as surrogate marker of HPV status and does not have any influence on our results.
Additionally, as the TNM used was Pathological Classification. Therefore, in Table 3, we changed the T and N to pT and pN, in order to make the table more accurate.
The paper by Ohashi et al. 2017 https://doi.org/10.1111/cas.13244 should be cited, since some results go in pairs : M2 macrophages, CD163 mRNA expression levels, HNSCC, ARG1. Please discuss.
In this paper The following statement is made in reference to their results and the citation to ref. 18. “HNSCC tissue had a significantly lower expression of ARG1 than that in normal pharyngeal tissue (Fig. S2). This result suggests that ARG1 is not an M2 macrophage marker in human tumor tissue.18“Please comment on this.
We carefully read the paper recommended by the reviewer. The paper from Ohashi et al. was additionally cited and discussed in our manuscript. However, it is important to note, that the conclusion “This result suggests that ARG1 is not an M2 macrophage marker in human tumor tissue.18” is problematic. It is known and noted in our manuscript, that ARG1 is also expressed by MDSCs and neutrophils, and this fact is not considered by Ohashi et al. in their paper. Since they used only qPCR analysis method for the detection of ARG1 on tumor tissue samples, they were not able to distinguish between these populations of cells. In our manuscript, ARG1 is detected on a protein level both on macrophages and non-macrophage population. Our qPCR data shows higher ARG1 expression in HPV- tumors, which is also supported by IHC. Similarly, ARG1 expression on human TAMs was observed by others, which suggests that ARG1 can be used as a M2 marker (not as an exclusive M2 marker):
Azambuja et al., (2020) https://doi.org/10.3390/ijms21113990
Massi et al., (2007) https://doi.org/10.1016/j.humpath.2007.02.018
Jiang et al., (2009) https://doi.org/10.1016/j.heliyon.2019.e02273
Benner et al., (2019) http://dx.doi.org/10.1186/s40425-019-0622-0
Shen et al., (2018) https://doi.org/10.2147/CMAR.S174899
As our study was focused on comparison of HPV+ and HPV- tumors, we did not evaluate TAMs markers in a healthy tissue. Ohashi et al. described lower ARG1 expression in HNSCC compared to pharyngeal tissue. It is in a conflict with Srivastava et al., which is cited in our manuscript.
A minor point is that in table 3, tumor size is indicated, but I assume the T stage from TNM is meant. In this case this is not correct, since t-stage may be dependent on infiltration on surrounding structures and not only size. Tumor extension for example would be more adequate.
We thank the reviewer for pointing this out. We agree and we corrected the expression of Tumor size to Tumor extension in Table 3, and in the text on lines 237 and 242.
Did the authors checked the ARG1 for the group that is HPV positive and (ex-)smoker? What were the results?
We thank the reviewer for his question. We analysed the ARG1 level in the groups with a respect to a smoking history. We observed the higher ARG1 level in the HPV+ group, but there were no differences of ARG1 level between smokers and non-smokers in the two groups when analysed based on HPV status of the tumors. As we did not observe any deaths in the HPV+ group of non-smokers, we were not able to perform the survival analyses.
